# Selecting Optimal Decisions via Distributionally Robust Nearest-Neighbor Regression

**Ruidi Chen**
Division of Systems Engineering
Boston University
Boston, MA 02215
rchen15@bu.edu

**Ioannis Ch. Paschalidis** [*]
Department of Electrical and Computer Engineering
Division of Systems Engineering
and Department of Biomedical Engineering
Boston University
Boston, MA 02215
yannisp@bu.edu

## Abstract

This paper develops a prediction-based prescriptive model for optimal decision making that (i) predicts the outcome under each action using a robust nonlinear model, and (ii) adopts a randomized prescriptive policy determined by the predicted outcomes. The predictive model combines a new regularized regression technique, which was developed using *Distributionally Robust Optimization (DRO)* with an ambiguity set constructed from the Wasserstein metric, with the *K-Nearest Neighbors (K-NN)* regression, which helps to capture the nonlinearity embedded in the data. We show theoretical results that guarantee the out-of-sample performance of the predictive model, and prove the optimality of the randomized policy in terms of the expected true future outcome. We demonstrate the proposed methodology on a hypertension dataset, showing that our prescribed treatment leads to a larger reduction in the systolic blood pressure compared to a series of alternatives. A clinically meaningful threshold level used to activate the randomized policy is also derived under a sub-Gaussian assumption on the predicted outcome.

## 1 Introduction

Suppose we are given a discrete set of available actions $[M] \triangleq \{1, \ldots, M\}$, and our goal is to choose $m \in [M]$ such that the future outcome $y$ is optimized. We are interested in finding the optimal decision with the aid of auxiliary data $\mathbf{x} \in \mathbb{R}^p$ that are concurrently observed, and correlated with the uncertain outcome $y$. A main challenge with learning from observational data lies in the lack of counterfactual information. One solution is to estimate/predict the effects of counterfactual policies by learning an action-dependent *predictive model* that groups the training samples based on their actions, and fits a model in each group between the outcome $y$ and the feature $\mathbf{x}$. The predictions from this composite model can be used to determine the optimal action to take. The performance of the prescribed decision hinges on the quality of the predictive model. We have observed that (i) there are often "outliers" in the data, especially in the medical applications motivating this work, caused by recording errors, missing values, and factors not captured in the data, and (ii) the underlying relationship we try to learn is usually nonlinear and its parametric form is not known a priori. To deal with these issues, a nonparametric robust learning procedure is in need.

Motivated by the observation that individuals with similar features $\mathbf{x}$ would have similar outcomes $y$ if they were to take the same action, we propose a predictive model that makes predictions based on the outcomes of similar individuals/*neighbors* in each group of the training set. It is a nonlinear and nonparametric estimator which constructs locally linear (constant) curves based on the similarity between individuals. To find reasonable neighbors, we need to accurately identify the set of features that are predictive of the outcome. We propose a regularized regression procedure for this task in

---

[*]http://sites.bu.edu/paschalidis

consideration of the outliers that could potentially bias the estimation. Our prescriptive methodology is established on the basis of a regression informed *K-Nearest Neighbors (K-NN)* model [2] that evaluates the importance of features through regularized regression, and estimates the outcome by averaging over the neighbors identified by a regression coefficients-weighted distance metric.

The regularized regression has its root in *Distributionally Robust Linear Regression (DRLR)* with a Wasserstein metric-based uncertainty set [13]. The K-NN model builds locally linear (and globally nonlinear) predictions using information from neighbors, accounting for the non-linearity that is not captured by DRLR. Furthermore, it is easy to estimate and is efficient to solve. Our framework uses both parametric (DRLR) and nonparametric (K-NN) predictive models, producing robust predictions immunized against outliers and capturing the underlying non-linearity in the data. It is more information-efficient and more interpretable than the vanilla K-NN. We then develop a randomized prescriptive policy that chooses each action $m$, whose predicted outcome is $\hat{y}_m(\mathbf{x})$, with probability $e^{-\xi \hat{y}_m(\mathbf{x})} / \sum_{j=1}^{M} e^{-\xi \hat{y}_j(\mathbf{x})}$, for some pre-specified positive constant $\xi$. As we will see, this randomized strategy leads to a nearly optimal future outcome by an appropriate choice of $\xi$.

In recent years there has been an emerging interest in combining ideas from machine learning with operations research to develop a framework that uses data to prescribe optimal decisions [4, 16, 10, 22]. Current research has focused on applying machine learning methodologies to predict the counterfactuals, based on which optimal decisions can be made. Local learning methods such as K-NN [2], LOESS (LOcally Estimated Scatterplot Smoothing) [15], CART (Classification And Regression Trees) [12], and Random Forests [11] have been studied [4, 7, 8, 17, 9]. Extensions to continuous and multi-dimensional decision spaces with observational data were considered in [5]. To prevent overfitting, [6] proposed two robust prescriptive methods based on Nadaraya-Watson and nearest-neighbors learning. Deviating from such a predict-optimize paradigm, [3] presented a new bandit algorithm based on the LASSO to learn a model of decision rewards conditional on individual-specific covariates.

Our method constructs a locally linear estimator of the future outcome through learning a robust metric in the feature space. Different from the classical metric learning works (e.g., [20]), we solve a downstream decision making problem by utilizing the information filtered by the learned metric. [20] focuses on the computational aspect of solving the metric regression problem. By contrast, we focus on developing a novel method for the optimal decision making problem rather than improving the algorithmic efficiency. Moreover, [20] studies only the regression problem, whereas we considered a richer framework of combining regression with a randomized prescriptive policy.

Our problem is closely related to contextual bandits [14, 1, 23, 25] where an agent learns a sequence of decisions conditional on the contexts, with the aim of maximizing its cumulative reward. It has recently found applications in learning personalized treatment for chronic diseases from mobile health data [24, 26, 27]. However, in this work, we learn the interaction between the context and rewards in each action group across similar individuals, not over the history of the same individual as in contextual bandits. Contextual bandits are most suitable for learning sequential strategies through repeated interactions with the environment, which requires a substantial amount of historical data for exploring the reward function and exploiting the promising actions. By contrast, our method does not require the availability of historical data, but instead learns the payoff function from similar individuals. This can be viewed as a different type of exploration, i.e., when little information can be acquired for the past states of an individual, investigating the behavior of similar subjects may be beneficial. This is essential for learning from Electronic Health Records (EHRs) available in the hospital, which do not include frequent patient data. For instance, we may observe a very sparse treatment history for some patients, and the lag between patient visits is usually large.

Our method is similar to K-NN regression with an *Ordinary Least Squares (OLS)*-weighted metric used in [7] to learn the optimal treatment for type-2 diabetic patients. The key differences lie in that: (i) we adopt a robustified regression procedure that is immunized against outliers and is thus more stable and reliable; (ii) we propose a randomized prescriptive policy that adds robustness to the methodology, whereas [7] deterministically prescribed the treatment with the best predicted outcome; (iii) we establish theoretical guarantees on the quality of the predictions and the prescribed actions, and (iv) the prescriptive rule in [7] was activated when the improvement of the recommended treatment over the standard of care exceeded a certain threshold whereas our method looks into the improvement over the previous regimen. This distinction makes our algorithm applicable in the scenario where the standard of care is unknown or ambiguous. Further, we derive a closed-form

expression for the threshold level, which greatly improves the computational efficiency compared to [7] where a threshold was selected by cross-validation.

The remainder of the paper is organized as follows. In Sec. 2, we introduce the DRLR+K-NN model and present the performance guarantees on its predictive power. Sec. 3 develops the randomized prescriptive policy and proves its optimality in terms of the expected true outcome. Experimental results using real medical (EHR) data are presented in Sec. 4. We conclude the paper in Sec. 5.

## 2 DRLR Informed K-Nearest Neighbors

Given a feature vector $\mathbf{x} \in \mathbb{R}^p$, and a set of $M$ available actions $[M]$, our goal is to predict the future outcome $y_m(\mathbf{x})$ under each possible action $m \in [M]$. Assume the following relationship between the features and the outcome:

$$y_m = \mathbf{x}_m' \boldsymbol{\beta}_m^* + h_m(\mathbf{x}_m) + \epsilon_m,$$

where prime denotes transpose, $(\mathbf{x}_m, y_m)$ represents the feature-outcome pair of an individual taking action $m$; $\boldsymbol{\beta}_m^*$ is the coefficient that captures the linear trend; $h_m(\cdot)$ is a Lipschitz continuous nonlinear function (whose form is unknown) describing the nonlinear fluctuation in $y_m$, and $\epsilon_m$ is the noise term with zero mean and standard deviation $\eta_m$ that expresses the intrinsic randomness of $y_m$ and is assumed to be independent of $\mathbf{x}_m$.

Suppose for each $m \in [M]$, we observe $N_m$ training samples $(\mathbf{x}_{mi}, y_{mi}), i = 1, \ldots, N_m$, that take action $m$. To estimate $\boldsymbol{\beta}_m^*$, we adopt the robust formulation that was developed in [13]. A robust model could lead to an improved out-of-sample performance, and accommodate the potential nonlinearity that is not explicitly revealed by the linear coefficients $\boldsymbol{\beta}_m^*$, thus resulting in a more accurate assessment of the features. The DRLR model developed in [13] minimizes the worst-case absolute loss within a distributional ambiguity set defined using the Wasserstein metric [18, 19] that contains all possible perturbations on the distribution of the data. The robustness is achieved through hedging against this family of distributions. The learning problem is formulated as:

$$\inf_{\boldsymbol{\beta}_m} \sup_{\mathbb{Q}_m \in \Omega_m} \mathbb{E}^{\mathbb{Q}_m} \big[ |y_m - \mathbf{x}_m' \boldsymbol{\beta}_m| \big], \tag{1}$$

where $\mathbb{Q}_m$ is the probability distribution of $(\mathbf{x}_m, y_m)$, belonging to some set $\Omega_m$ defined as:

$$\Omega_m \triangleq \{ \mathbb{Q}_m \in \mathcal{M}(\mathcal{Z}_m) : W_1(\mathbb{Q}_m, \hat{\mathbb{P}}_{N_m}) \leq r_m \},$$

where $\mathcal{Z}_m$ is the set of all possible values for $(\mathbf{x}_m, y_m)$; $\mathcal{M}(\mathcal{Z}_m)$ is the space of probability distributions supported on $\mathcal{Z}_m$; $\hat{\mathbb{P}}_{N_m}$ is the uniform empirical distribution on the $N_m$ observed samples $(\mathbf{x}_{mi}, y_{mi}), i = 1, \ldots, N_m$; $r_m$ is a pre-specified parameter indicating the amount of ambiguity allowed; and $W_1(\mathbb{Q}_m, \hat{\mathbb{P}}_{N_m})$ is the order-1 Wasserstein distance between $\mathbb{Q}_m$ and $\hat{\mathbb{P}}_{N_m}$ defined as:

$$W_1(\mathbb{Q}_m, \hat{\mathbb{P}}_{N_m}) = \sup_{f \in \mathcal{L}} \left\{ \int_{\mathcal{Z}_m} f(\mathbf{z}_m) \, \mathbb{Q}_m(d\mathbf{z}_m) - \int_{\mathcal{Z}_m} f(\mathbf{z}_m) \, \hat{\mathbb{P}}_{N_m}(d\mathbf{z}_m) \right\},$$

where $\mathbf{z}_m = (\mathbf{x}_m, y_m)$, and $\mathcal{L}$ is the space of all Lipschitz continuous functions satisfying $|f(\mathbf{z}_{m1}) - f(\mathbf{z}_{m2})| \leq \|\mathbf{z}_{m1} - \mathbf{z}_{m2}\|_2, \forall \mathbf{z}_{m1}, \mathbf{z}_{m2} \in \mathcal{Z}_m$.

With $N_m$ independently and identically distributed samples $(\mathbf{x}_{mi}, y_{mi}), i = 1, \ldots, N_m$, [13] has shown that problem (1) can be reformulated as:

$$\inf_{\boldsymbol{\beta}_m} \frac{1}{N_m} \sum_{i=1}^{N_m} |y_{mi} - \mathbf{x}_{mi}' \boldsymbol{\beta}_m| + r_m \|(-\boldsymbol{\beta}_m, 1)\|_2. \tag{2}$$

Solving Eq. (2) gives us a robust estimator of the linear regression coefficient $\boldsymbol{\beta}_m^*$, which we denote by $\hat{\boldsymbol{\beta}}_m$. The elements of $\hat{\boldsymbol{\beta}}_m$ measure the relative significance of the predictors in determining the outcome $y_m$. We feed the estimator into the nonlinear non-parametric K-NN regression model, by considering the following $\hat{\boldsymbol{\beta}}_m$-weighted metric:

$$\|\mathbf{x} - \mathbf{x}_{mi}\|_{\hat{\mathbf{W}}_m} = \sqrt{(\mathbf{x} - \mathbf{x}_{mi})' \hat{\mathbf{W}}_m (\mathbf{x} - \mathbf{x}_{mi})}, \tag{3}$$

where $\hat{\mathbf{W}}_m$ is a diagonal matrix with elements $(\hat{\beta}_{m1})^2, \ldots, (\hat{\beta}_{mp})^2$, with $\hat{\beta}_{mi}$ the $i$-th element of $\hat{\boldsymbol{\beta}}_m$. For a new test sample $\mathbf{x}$, within each group $m$, we find its $K_m$ nearest neighbors using the weighted distance function (3). The predicted future outcome for $\mathbf{x}$ under action $m$, denoted by $\hat{y}_m(\mathbf{x})$, is computed as the average response among the $K_m$ nearest neighbors, i.e.,

$$\hat{y}_m(\mathbf{x}) = \frac{1}{K_m} \sum_{i=1}^{K_m} y_{m(i)}, \tag{4}$$

where $y_{m(i)}$ is the response of the $i$-th closest sample to $\mathbf{x}$ within group $m$. Eq. (4) computes a K-NN estimate of the future outcome by using the linear regression coefficients weighted distance function, which can be viewed as a locally smoothed estimator in the neighborhood of $\mathbf{x}$. The intuition for using the $\hat{\boldsymbol{\beta}}_m$-weighted metric is to amplify the weight of features that are most predictive of $y_m$ and downweight the unimportant ones. As a result, the selected samples are close to $\mathbf{x}$ in the most relevant features, and their corresponding response values should serve as a good approximation. Notice that Eq. (4) treats all neighbors equally by using the same weight. An alternative is to take a distance-weighted average of the responses of neighbors; we have numerically tried this strategy on our medical datasets in Section 4, but we find that its effect is not significantly different from the strategy where a uniform average of the responses is taken. We also want to point out that the following theoretical analysis can be easily adapted to the weighted average response prediction.

We next show that Eq. (4) provides a good prediction in the sense of *Mean Squared Error (MSE)*. The bias-variance decomposition implies the following:

$$
\begin{aligned}
\mathrm{MSE}&\big(\hat{y}_m(\mathbf{x}) \big| \mathbf{x}, \mathbf{x}_{mi}, i=1,\ldots,N_m\big) \triangleq \mathbb{E}\Big[\big(\hat{y}_m(\mathbf{x}) - y_m(\mathbf{x})\big)^2 \Big| \mathbf{x}, \mathbf{x}_{mi}, i=1,\ldots,N_m\Big] \\
&= \mathbb{E}\Bigg[\Big(\frac{1}{K_m}\sum_{i=1}^{K_m}\big(\mathbf{x}'_{m(i)}\boldsymbol{\beta}^*_m + h_m(\mathbf{x}_{m(i)}) + \epsilon_{m(i)}\big) - \big(\mathbf{x}'\boldsymbol{\beta}^*_m + h_m(\mathbf{x}) + \epsilon_m\big)\Big)^2 \Big| \mathbf{x}, \mathbf{x}_{mi}, \forall i\Bigg] \\
&= \Big(\mathbf{x}'\boldsymbol{\beta}^*_m + h_m(\mathbf{x}) - \frac{1}{K_m}\sum_{i=1}^{K_m}\big(\mathbf{x}'_{m(i)}\boldsymbol{\beta}^*_m + h_m(\mathbf{x}_{m(i)})\big)\Big)^2 + \frac{\eta^2_m}{K_m} + \eta^2_m \\
&= \Big(\frac{1}{K_m}\sum_{i=1}^{K_m}\big((\mathbf{x} - \mathbf{x}_{m(i)})'\boldsymbol{\beta}^*_m + h_m(\mathbf{x}) - h_m(\mathbf{x}_{m(i)})\big)\Big)^2 + \frac{\eta^2_m}{K_m} + \eta^2_m,
\end{aligned} \tag{5}
$$

where $y_m(\mathbf{x})$ is the *true* future outcome on $\mathbf{x}$ if action $m$ is prescribed; and $\mathbf{x}_{m(i)}$, $\epsilon_{m(i)}$ are the feature vector and the noise term corresponding to the $i$-th closest sample to $\mathbf{x}$ within group $m$, respectively. The third equality comes from the fact that the error term is independent of the features. For each $m \in [M]$, we aim to provide a probabilistic bound for 5 w.r.t. the measure of the $N_m$ training samples. By examining the first term of the last line of (5), we see that for MSE to be small, the following three conditions need to be satisfied: (i) $\|\boldsymbol{\beta}^*_m - \hat{\boldsymbol{\beta}}_m\|_2$ is small; (ii) $\|\mathbf{x} - \mathbf{x}_{m(i)}\|_{\hat{\mathbf{W}}_m}$ is small for $i = 1, \ldots, K_m$; and (iii) $h_m(\mathbf{x}) - h_m(\mathbf{x}_{m(i)})$ is small for $i = 1, \ldots, K_m$. Below we state the assumptions that are needed to establish the result.

**Assumption A** $\|(\mathbf{x}_m, y_m)\|_2 \leq R_m$ *a.s.*.

**Assumption B** $\|(-\boldsymbol{\beta}_m, 1)\|_2 \leq \bar{B}_m$.

**Assumption C** *For some set $\mathcal{A}(\boldsymbol{\beta}^*_m) = cone\{\mathbf{v}| \ \|(-\boldsymbol{\beta}^*_m, 1) + \mathbf{v}\|_2 \leq \|(-\boldsymbol{\beta}^*_m, 1)\|_2\} \cap \mathbb{S}^{p+1}$ and some positive scalar $\underline{\alpha}_m$, where $\mathbb{S}^{p+1}$ is the unit sphere in the $(p+1)$-dimensional Euclidean space,*

$$\inf_{\mathbf{v} \in \mathcal{A}(\boldsymbol{\beta}^*_m)} \mathbf{v}' \mathbf{Z}_m \mathbf{Z}'_m \mathbf{v} \geq \underline{\alpha}_m,$$

*where $\mathbf{Z}_m = [\mathbf{z}_{m1} \cdots \mathbf{z}_{mN_m}]$ is the matrix with columns $\mathbf{z}_{m1}, \ldots, \mathbf{z}_{mN_m}$, with $\mathbf{z}_{mi} = (\mathbf{x}_{mi}, y_{mi})$.*

**Assumption D** $(\mathbf{x}_m, y_m)$ *is a centered sub-Gaussian random vector, i.e., it has zero mean and satisfies the following condition:*

$$\||(\mathbf{x}_m, y_m)\||_{\psi_2} = \sup_{\mathbf{u} \in \mathbb{S}^{p+1}} \||(\mathbf{x}_m, y_m)'\mathbf{u}\||_{\psi_2} \leq \mu_m.$$

**Assumption E** *The covariance matrix of $(\mathbf{x}_m, y_m)$ has bounded positive eigenvalues. Set $\mathbf{\Gamma}_m = \mathbb{E}[(\mathbf{x}_m, y_m)(\mathbf{x}_m, y_m)']$; then,*

$$0 < \lambda_{m0} \triangleq \lambda_{min}(\mathbf{\Gamma}_m) \leq \lambda_{max}(\mathbf{\Gamma}_m) \triangleq \lambda_{m1} < \infty.$$

To see the validity of the above assumptions, notice that with standardized data, Assumptions A and B are easily satisfied. Assumptions C, D, and E bound the variance of $(\mathbf{x}, y)$ in terms of the eigenvalues of its covariance matrix and its sub-Gaussian norm. (If the variance in the data is prohibitively high, the samples would contain little information to learn from.) Due to limited space, we defer the intermediate results that bound $\|\boldsymbol{\beta}_m^* - \hat{\boldsymbol{\beta}}_m\|_2$ and $\|\mathbf{x} - \mathbf{x}_{m(i)}\|_{\hat{\mathbf{W}}_m}$ to the supplementary. But those results will be used as the foundation to derive the bound on the MSE of $\hat{y}_m(\mathbf{x})$.

**Theorem 2.1** *Suppose we are given $N_m$ i.i.d. copies of $(\mathbf{x}_m, y_m)$, denoted by $(\mathbf{x}_{mi}, y_{mi}), i = 1, \ldots, N_m$, where $\mathbf{x}_m$ has independent, centered coordinates, and*

$$cov(\mathbf{x}_m) = diag\left(\sigma_{m1}^2, \ldots, \sigma_{mp}^2\right).$$

*Given a fixed predictor $\mathbf{x} = (x_1, \ldots, x_p)$, and some scalar $\bar{w}_m$, assuming*

1. *$h_m(\cdot)$ is Lipschitz continuous with a Lipschitz constant $L_m$ on the metric spaces $(\mathcal{X}_m, \|\cdot\|_2)$ and $(\mathcal{Y}_m, |\cdot|)$, where $\mathcal{X}_m, \mathcal{Y}_m$ are the domain and codomain of $h_m(\cdot)$, respectively.*

2. *$\bar{w}_m^2 > \bar{B}_m^2 \sum_{j=1}^p (\sigma_{mj}^2 + x_j^2)$, where $\bar{B}_m$ is specified in Assumption B.*

3. *$|(x_{mij} - x_j)^2 - (\sigma_{mj}^2 + x_j^2)| \leq T_m, \forall i, j$, where $x_{mij}$ is the $j$-th component of $\mathbf{x}_{mi}$.*

4. *The coordinates of any feasible solution to (2) have absolute values greater than or equal to some positive number $b_m$ (dense estimators).*

*Under Assumptions A, B, C, D, E, when $N_m \geq n_m$, with probability at least $\delta_m - I_{1-p_{m0}}(N_m - K_m + 1, K_m)$ w.r.t. the measure of samples,*

$$\mathbb{E}\left[(\hat{y}_m(\mathbf{x}) - y_m(\mathbf{x}))^2 \Big| \mathbf{x}, \mathbf{x}_{mi}, i = 1, \ldots, N_m\right] \leq \left(\frac{\bar{w}_m \tau_m}{b_m} + \sqrt{p}\bar{w}_m + \frac{L_m \bar{w}_m}{\bar{B}_m}\right)^2 + \frac{\eta_m^2}{K_m} + \eta_m^2, \quad (6)$$

*and for any $a \geq \left(\frac{\bar{w}_m \tau_m}{b_m} + \sqrt{p}\bar{w}_m + \frac{L_m \bar{w}_m}{\bar{B}_m}\right)^2 + \frac{\eta_m^2}{K_m} + \eta_m^2$,*

$$\mathbb{P}\left((\hat{y}_m(\mathbf{x}) - y_m(\mathbf{x}))^2 \geq a \Big| \mathbf{x}, \mathbf{x}_{mi}, i = 1, \ldots, N_m\right) \leq \frac{\left(\frac{\bar{w}_m \tau_m}{b_m} + \sqrt{p}\bar{w}_m + \frac{L_m \bar{w}_m}{\bar{B}_m}\right)^2 + \frac{\eta_m^2}{K_m} + \eta_m^2}{a},$$
$$(7)$$

*where $I_{1-p_{m0}}(\cdot, \cdot)$ is the regularized incomplete beta function, and*

$$p_{m0} = 1 - \exp\left(-\frac{\sigma_m^2}{T_m^2} g\left(\frac{T_m\left(\bar{w}_m^2 / \bar{B}_m^2 - \sum_j (\sigma_{mj}^2 + x_j^2)\right)}{\sigma_m^2}\right)\right),$$

*with*

$$\sigma_m = \sqrt{\sum_{j=1}^p var\left((x_{mij} - x_j)^2\right)}, \qquad g(u) = (1 + u)\log(1 + u) - u.$$

*The notations $n_m, \delta_m$, and $\tau_m$ come from a simplified version of Theorem 3.11 in [13], which states that when the sample size $N_m \geq n_m$, with probability at least $\delta_m$,*

$$\|\boldsymbol{\beta}_m^* - \hat{\boldsymbol{\beta}}_m\|_2 \leq \tau_m.$$

*The parameters $n_m, \delta_m, \tau_m$ are related to the sub-Gaussian norm of $(\mathbf{x}_m, y_m)$, the eigenvalues of the covariance matrix of $(\mathbf{x}_m, y_m)$, and the geometric structure of the true regression coefficient $\boldsymbol{\beta}_m^*$.*

**Remark 2.1** The expectation in (6) and the probability in (7) are w.r.t. the measure of the noise $\epsilon_m$. Thm. 2.1 essentially says that for any given $\mathbf{x}$, with a high probability (w.r.t. the measure of samples), the prediction is close to the true future outcome. The prediction bias is determined by the accuracy

of the linear coefficient estimate, the similarity between the individual in query and its $K$ nearest neighbors, the dimensionality of data, and the smoothness of the regression hypothesis.

**Remark 2.2** The dependence on $b_m$ in the upper bound provided by (6) is due to the fact that $\hat{\mathbf{W}}_m$ has diagonal elements $\hat{\beta}_{mj}^2, j = 1, \ldots, p$, which are assumed to be at least $b_m^2$. If we multiply $\hat{\mathbf{W}}_m$ by a very large number, the neighbor selection criterion is not affected, since the relative significance of the predictors stays unchanged, but the $b_m$ appearing in (6) would be replaced by a very large number, diminishing the effect of the first term in the parenthesis, at the price of increasing $\check{B}_m$ and $\bar{w}_m$, which in turn has an effect on the number of neighbors that are needed. It might be interesting to explore this implicit trade-off and find the optimal $\hat{\mathbf{W}}_m$ to achieve the smallest MSE. For simplicity, we just use $\hat{\mathbf{W}}_m = \text{diag}(\hat{\beta}_{m1}^2, \ldots, \hat{\beta}_{mp}^2)$ in this work.

**Remark 2.3** We offered similar insights to [20] for the generalization bounds. Theorem 5.1 in [20] provided a risk bound that depends on the empirical risk (reflected in $\tau_m$ and $\bar{w}_m$ of our bound), the dimensionality of data ($p$), and the smoothness of the regression hypothesis ($L_m$).

## 3   Prescriptive Policy Development

We now proceed to develop the prescriptive policy with the aim of minimizing the future outcome. A natural idea is to take the action that yields the minimum predicted outcome. To allow for flexibility in exploring alternatives that have a comparable performance, and also to correct for potential prediction errors that might mislead the ranking of actions, we propose a randomized policy that prescribes each action with a probability inversely proportional to its exponentiated predicted outcome. It can be viewed as an offline Hedge algorithm [21] that increases the robustness of our method through exploration. Specifically, given an individual with a feature vector $\mathbf{x}$, and her predicted future outcome under each action $m$, denoted by $\hat{y}_m(\mathbf{x})$, we consider a randomized policy that chooses action $m$ with probability $e^{-\xi \hat{y}_m(\mathbf{x})} / \sum_{j=1}^{M} e^{-\xi \hat{y}_j(\mathbf{x})}$, with $\xi$ being some pre-specified positive constant. We would like to explore properties of this policy in terms of its expected *true* outcome.

**Theorem 3.1** *Given any fixed predictor* $\mathbf{x} \in \mathbb{R}^p$, *denote its predicted and true future outcome under action* $m$ *by* $\hat{y}_m(\mathbf{x})$ *and* $y_m(\mathbf{x})$, *respectively. Assume that we adopt a randomized strategy that prescribes action* $m$ *with probability* $e^{-\xi \hat{y}_m(\mathbf{x})} / \sum_{j=1}^{M} e^{-\xi \hat{y}_j(\mathbf{x})}$, *for some* $\xi \geq 0$. *Assume* $\hat{y}_m(\mathbf{x})$ *and* $y_m(\mathbf{x})$ *are non-negative,* $\forall m$. *For any* $k \in [M]$, *the expected true outcome of this policy satisfies:*

$$
\begin{aligned}
\sum_{m=1}^{M} \frac{e^{-\xi \hat{y}_m(\mathbf{x})}}{\sum_j e^{-\xi \hat{y}_j(\mathbf{x})}} y_m(\mathbf{x}) \leq{} & y_k(\mathbf{x}) + \left( \hat{y}_k(\mathbf{x}) - \frac{1}{M} \sum_{m=1}^{M} \hat{y}_m(\mathbf{x}) \right) \\
& + \xi \left( \frac{1}{M} \sum_{m=1}^{M} \hat{y}_m^2(\mathbf{x}) + \sum_{m=1}^{M} \frac{e^{-\xi \hat{y}_m(\mathbf{x})}}{\sum_j e^{-\xi \hat{y}_j(\mathbf{x})}} y_m^2(\mathbf{x}) \right) + \frac{\log M}{\xi}.
\end{aligned}
\tag{8}
$$

Theorem 3.1 says that the expected *true* outcome of the randomized policy is no worse than the true outcome of any action $k$ plus two components, one accounting for the gap between the *predicted* outcome under $k$ and the average predicted outcome, and the other depending on the parameter $\xi$. Thinking about choosing $k = \arg\min_m y_m(\mathbf{x})$, if $\hat{y}_k(\mathbf{x})$ is below the average predicted outcome (which should be true if we have an accurate prediction), it follows from (8) that the randomized policy leads to a nearly optimal future outcome by an appropriate choice of $\xi$.

In medical applications, when determining the *future* prescription for a patient, we usually have access to some auxiliary information such as the *current* prescription she is receiving, and her *current* measurements. In consideration of the health care costs and treatment transients, it is not desired to switch patients' treatments too frequently. We thus set a threshold level for the expected improvement in the outcome, below which the randomized strategy will be rendered inactive and the current therapy will be continued. Specifically, if $\sum_k \frac{e^{-\xi \hat{y}_k(\mathbf{x})}}{\sum_j e^{-\xi \hat{y}_j(\mathbf{x})}} \hat{y}_k(\mathbf{x}) \leq x_{\text{co}} - T(\mathbf{x})$, $m_{\text{f}}(\mathbf{x}) = m$ w.p. $e^{-\xi \hat{y}_m(\mathbf{x})} / \sum_{j=1}^{M} e^{-\xi \hat{y}_j(\mathbf{x})}$; otherwise $m_{\text{f}}(\mathbf{x}) = m_{\text{c}}(\mathbf{x})$, where $m_{\text{f}}(\mathbf{x})$ and $m_{\text{c}}(\mathbf{x})$ are the future and current prescriptions for patient $\mathbf{x}$, respectively; $m$ is the prescribed action under the randomized policy; $x_{\text{co}}$ represents the current observed outcome (e.g., current blood pressure), which is assumed to be one of the components of $\mathbf{x}$; and $T(\mathbf{x})$ is some threshold level which will be determined later. This prescriptive rule basically says that the randomized strategy will be activated only if the expected improvement relative to the current observed outcome is significant.

**Theorem 3.2** *Assume that the distribution of the predicted outcome $\hat{y}_m(\mathbf{x})$ conditional on $\mathbf{x}$, is sub-Gaussian, and its $\psi_2$-norm is equal to $\sqrt{2}C_m(\mathbf{x})$, for any $m \in [M]$ and any $\mathbf{x}$. Given a small $0 < \bar{\epsilon} < 1$, in order to satisfy*

$$\mathbb{P}\left(\sum_k \frac{e^{-\xi \hat{y}_k(\mathbf{x})}}{\sum_j e^{-\xi \hat{y}_j(\mathbf{x})}} \hat{y}_k(\mathbf{x}) > x_{co} - T(\mathbf{x})\right) \leq \bar{\epsilon},$$

*it suffices to set a threshold*

$$T(\mathbf{x}) = \max\left(0, \ \min_m \left(x_{co} - \mu_{\hat{y}_m}(\mathbf{x}) - \sqrt{-2C_m^2(\mathbf{x}) \log(\bar{\epsilon}/M)}\right)\right),$$

*where $\mu_{\hat{y}_m}(\mathbf{x}) = \mathbb{E}[\hat{y}_m(\mathbf{x})|\mathbf{x}]$.*

Theorem 3.2 finds the largest threshold $T(\mathbf{x})$ such that the probability of the expected improvement being less than $T(\mathbf{x})$ is small. The parameters $\mu_{\hat{y}_m}(\mathbf{x})$ and $C_m(\mathbf{x})$ for $m \in [M]$ can be estimated by simulation through random sampling from a subset of the training examples.

---

**Algorithm 1** Estimating the conditional mean and standard deviation of the predicted outcome.

---

**Input:** a feature vector $\mathbf{x}$; $a_m$: the number of subsamples used to compute $\hat{\boldsymbol{\beta}}_m$, $a_m < N_m$; $d_m$: the number of repetitions.

**for** $i = 1, \ldots, d_m$ **do**

Randomly pick $a_m$ samples from group $m$, and use them to estimate a robust regression coefficient $\hat{\boldsymbol{\beta}}_{m_i}$ through solving 2.

The future outcome for $\mathbf{x}$ under action $m$ is predicted as $\hat{y}_{m_i}(\mathbf{x}) = \mathbf{x}'\hat{\boldsymbol{\beta}}_{m_i}$.

**end for**

**Output:** Estimate the conditional mean of $\hat{y}_m(\mathbf{x})$ as:

$$\mu_{\hat{y}_m}(\mathbf{x}) = \frac{1}{d_m}\sum_{i=1}^{d_m}\hat{y}_{m_i}(\mathbf{x}),$$

and the conditional standard deviation as:

$$C_m(\mathbf{x}) = \sqrt{\frac{1}{d_m - 1}\sum_{i=1}^{d_m}\left(\hat{y}_{m_i}(\mathbf{x}) - \mu_{\hat{y}_m}(\mathbf{x})\right)^2}.$$

---

**A Special Case.** As $\xi \to \infty$, the randomized policy will assign probability 1 to the action with the lowest predicted outcome, which is equivalent to the following deterministic policy:

$$m_{\mathrm{f}}(\mathbf{x}) = \begin{cases} \arg\min_m \hat{y}_m(\mathbf{x}), & \text{if } \min_m \hat{y}_m(\mathbf{x}) \leq x_{\mathrm{co}} - T(\mathbf{x}), \\ m_{\mathrm{c}}(\mathbf{x}), & \text{otherwise.} \end{cases}$$

A slight modification to the threshold level $T(\mathbf{x})$ is given below:

$$T(\mathbf{x}) = \max\left(0, \ \min_m \left(x_{\mathrm{co}} - \mu_{\hat{y}_m}(\mathbf{x}) - \sqrt{-2C_m^2(\mathbf{x}) \log \bar{\epsilon}}\right)\right).$$

# 4   Numerical Results on a Hypertension Dataset

In this section, we will apply our method to develop optimal prescriptions for patients with hypertension. The data used for the study come from a large academic hospital system handling more than 1 million patient visits per year and consist of *Electronic Health Records (EHR)* containing the patients' medical history in the period 1999–2014. Our goal is to find the treatment that minimizes the future systolic blood pressure based on the medical histories.

## 4.1 Dataset Description

According to certain cohort selection criteria (see the supplementary), we have identified 49,401 patients who have been diagnosed with hypertension. Each patient may have multiple entries in her/his medical record. To capture the period when the patient was experiencing the effect of the drug regimen, we define the *line of therapy* as a time period (between 200 and 500 days) during which the combination of drugs prescribed to the patient does not change.

**Patient Visits.** During each line of therapy, we split the treatment history of each patient into several *patient visits*, to reflect changes in the features and outcomes. The patient visits are considered to be occurring every 70 days. The measurements and lab tests are averaged over the 10 days prior to the visit. We define the *current prescription* of each visit as the combination of drugs that were given during the 10 days immediately preceding the visit, and the *standard of care* as the drug regimen that is prescribed by the doctors at the time of the visit. The *future* outcome of the visit is computed as the average systolic blood pressure in mmHg 70 to 180 days after it. Patient visits that contain missing values for the outcome are dropped. We have obtained 26,128 valid visits, each with 63 features.

**Features.** The features for building the predictive model include: (i) *demographics*: sex, age and race; (ii) *measurements*: systolic blood pressure and diastolic blood pressure, Body Mass Index (BMI) and pulse; (iii) *lab tests*: blood chemistry tests and hematology tests; and (iv) *diagnosis history*.

**Prescriptions.** We consider six types of prescriptions for hypertension, each corresponding to a different medication that could be prescribed: ACE inhibitor, Angiotensin Receptor Blockers (ARB), calcium channel blockers, thiazide and thiazide-like diuretics, $\alpha$-blockers and $\beta$-blockers. The patient visits are grouped based on their standard of care.

## 4.2 Model Development and Results

We will compare our algorithm with several alternatives that replace our DRLR informed K-NN with a different predictive model such as LASSO, CART, and OLS informed K-NN [7]. Both deterministic and randomized prescriptive policies are considered using predictions from these models.

**Parameter tuning.** Within each prescription group, we randomly split the patient visits into three sets: a training set (80%), a validation set (10%), and a test set (10%). To reflect the dependency of the number of neighbors on the number of training samples, we perform a linear regression between these two quantities, which we use to determine the number of neighbors needed in different settings. To tune the exponent $\xi$ for the randomized strategy, we need to evaluate the effects of counterfactual treatments. We assess the predictive power of a series of robust predictive models (see the supplementary) in terms of their $R^2$ and out-of-sample estimation errors, and select the DRLR+K-NN model (*imputation model*) that excels in all metrics, to impute the outcome for an unobservable treatment $m$, using the validation set.

When comparing the predictive performance of the models (Table 1, Supplementary), we fit a common regression model to all patients without dividing them into groups, with the prescription being used as a predictor. This leads to a significant reduction in the unexplained variance of $y$, and thus the advantages of DRLR+K-NN are not significant. However, when we do groupwise regression where prescription is not used as a predictor, the unexplained noise increases, robustness becomes more critical, and thus the advantages of our method become more prominent (see the following Table 1).

**Model training.** We solve the predictive models on the whole training set with the best tuned parameters, the output of which is used to develop the optimal prescriptions for the test set patients. The parameter $\bar{\epsilon}$ in the threshold $T(\mathbf{x})$ is set to $0.1$. We compute the average improvement (reduction) in outcomes for patients in the test set, which is defined to be the difference between the (expected) *future* outcome under the recommended therapy and the *current* observed outcome. If the recommendation does not match the standard of care, its future outcome is estimated through the imputation model that was discussed earlier, where $K_m$ should be selected to fit the size of the *test set*.

**Refinement of the policy.** In consideration of the sensitivity of K-NN to the number of neighbors, we propose a refinement of our model, where a patient-specific number of neighbors $K_m$ is used,

and the neighbors that are relatively far away from the patient in query are discarded. This can be considered as taking a weighted average of the responses of neighbors to make the K-NN prediction. Specifically, denote by $d_i^m$ the distance between the patient in query and her $i$-th closest neighbor in group $m$; we know $d_1^m \leq d_2^m \leq \ldots d_{K_m}^m$. Define $j_m^* = \arg\max_j \left( d_j^m - \sum_{i=1}^{j-1} \frac{d_i^m}{j-1} \right)$. Given some threshold $\tilde{T}$, the number of neighbors $K_m'$ will be determined as follows.

$$K_m' = \begin{cases} j_m^* - 1, \text{ if } \dfrac{d_{j_m^*}^m - \sum_{i=1}^{j_m^*-1} \frac{d_i^m}{j_m^*-1}}{\sum_{i=1}^{j_m^*-1} \frac{d_i^m}{j_m^*-1}} > \tilde{T}, \\ K_m, \quad \text{otherwise.} \end{cases}$$

**Results and discussion.** The reductions in outcomes (future minus current) for various models are shown in Table 1. The columns indicate the prescriptive policies (deterministic or randomized); the rows represent the predictive models whose outcomes $\hat{y}_m(\mathbf{x})$ serve as inputs to the prescriptive algorithm. We compare two strategies that use different rules for selecting the number of neighbors, with a validated threshold $\tilde{T} = 1$ for the patient-specific strategy. We test the performance of all algorithms over five repetitions, each with a different training set. We also list the reductions in outcomes resulting from the *standard of care*, and the *current prescription* which continues the current drug regimen.

Several observations are in order: (i) all models outperform the current prescription and the standard of care; (ii) the DRLR+K-NN model leads to the largest reduction in outcomes with a relatively stable performance; (iii) using a patient-specific $K_m'$ in general leads to a more significant reduction in outcomes, and (iv) the randomized policy achieves a similar (slightly better) performance than the deterministic one. Overall, the best DRLR+K-NN model leads to a 69% reduction in future systolic blood pressure compared to the 2nd best model. We expect the randomized strategy to win when the effects of several treatments do not differ much, in which case the deterministic algorithm might produce misleading results. The randomized policy could potentially improve the out-of-sample performance, as it gives the flexibility of exploring options that are suboptimal on the training set, but might be optimal on the test set.

It may be argued that the observed improvement is due to the evaluation model we choose; specifically, using DRLR+K-NN to assess the performance of all candidates might cause bias that favors our method. To mitigate this bias, we also used a mixture of OLS+K-NN and DRLR+K-NN (with equal weights) as the imputation model, given that they achieve the best predictive performance. Under this scheme, our model still outperforms all others.

Table 1: The reduction in future systolic blood pressured (mmHg); *mean (standard deviation).*

| | Training with a patient-specific $K_m'$ | | Training with a uniform $K_m$ | |
|---|---|---|---|---|
| | Deterministic | Randomized | Deterministic | Randomized |
| LASSO | -4.34 (0.28) | -4.33 (0.28) | -4.22 (0.20) | -4.22 (0.19) |
| CART | -4.46 (0.46) | -4.49 (0.50) | -4.48 (0.55) | -4.51 (0.49) |
| OLS+K-NN | -4.30 (0.35) | -4.30 (0.32) | -4.27 (0.32) | -4.29 (0.31) |
| DRLR+K-NN | -7.42 (0.46) | -7.58 (0.51) | -6.58 (0.70) | -6.78 (0.73) |
| Current prescription | -2.56 (0.14) | | -2.50 (0.16) | |
| Standard of care | -2.37 (0.11) | | -2.37 (0.11) | |

## 5   Conclusions

We developed a prediction-based prescriptive method that determines the probability of taking each action based on the predictions from a DRLR informed K-NN model. Theoretical guarantees on the out-of-sample performance of the predictive model and the optimality of the prescriptive algorithm were established. We also derived a closed-form expression for the threshold level that is used to activate the randomized policy. The proposed approach was applied to actual hypertension patient data obtained from a major academic hospital system, providing numerical evidence for the superiority of our algorithm in terms of the improvement in outcomes.

**Acknowledgments**

The research was partially supported by the NSF under grants IIS-1914792, DMS-1664644, and CNS-1645681, by the ONR under grant N00014-19-1-2571, by the NIH under grant 1R01GM135930, by the Boston University Data Science Initiative, and by the BU Center for Information and Systems Engineering.

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
