[Supplementary Material]

# Supplementary for Selecting Optimal Decisions via Distributionally Robust Nearest-Neighbor Regression

**Ruidi Chen**
Division of Systems Engineering
Boston University
Boston, MA 02215
rchen15@bu.edu

**Ioannis Ch. Paschalidis** *
Department of Electrical and Computer Engineering
Division of Systems Engineering
and Department of Biomedical Engineering
Boston University
Boston, MA 02215
yannisp@bu.edu

## A  Key Concepts

### A.1  Sub-Gaussian Random Variables

**Definition 1** (Sub-Gaussian random variable/vector). *A random variable $y \in \mathbb{R}$ with mean $\mu_y \triangleq \mathbb{E}(y)$ is sub-Gaussian if there exists some positive constant $C$ such that the tail of $y$ satisfies:*

$$\mathbb{P}(|y - \mu_y| \geq t) \leq 2\exp(-t^2/(2C^2)), \forall t \geq 0. \tag{1}$$

*The smallest constant $\sqrt{2}C$ satisfying (1) is called the sub-Gaussian norm, or the $\psi_2$-norm of $y$, denoted as $\|y\|_{\psi_2}$. All sub-Gaussian random variables have a finite $\psi_2$-norm. A random vector $\mathbf{z} \in \mathbb{R}^p$ is sub-Gaussian if $\mathbf{z}'\mathbf{u}$ is sub-Gaussian for any $\mathbf{u} \in \mathbb{R}^p$. The $\psi_2$-norm of a vector $\mathbf{z}$ is defined as:*

$$\|\mathbf{z}\|_{\psi_2} \triangleq \sup_{\mathbf{u} \in \mathbb{S}^p} \|\mathbf{z}'\mathbf{u}\|_{\psi_2},$$

*where $\mathbb{S}^p$ denotes the unit sphere in the $p$-dimensional Euclidean space.*

The sub-Gaussian property (1) describes a class of distributions whose tail decays at least as fast as a Gaussian; some classical examples include the Gaussian, Bernoulli, and any bounded distribution. An equivalent property to (1) says the following:

$$\mathbb{E}[\exp(\lambda y)] \leq \exp\left(\frac{\lambda^2 C^2}{2} + \lambda\mu_y\right), \qquad \forall \lambda \in \mathbb{R}.$$

The $\psi_2$-norm of a sub-Gaussian random variable is usually related to its standard deviation, and thus characterizes the random fluctuation embedded in the variable. For example, for a Gaussian random variable $y \sim \mathcal{N}(\mu_y, \sigma^2)$, its *Moment Generating Function (MGF)* is $M(\lambda) \triangleq \mathbb{E}[\exp(\lambda y)] = \exp(\lambda^2\sigma^2/2 + \lambda\mu_y)$, which implies that its $\psi_2$-norm is just a multiple of $\sigma$.

### A.2  Gaussian width

**Definition 2** (Gaussian width). *For any set $\mathcal{A} \subseteq \mathbb{R}^m$, its Gaussian width is defined as:*

$$w(\mathcal{A}) \triangleq \mathbb{E}\left[\sup_{\mathbf{u} \in \mathcal{A}} \mathbf{u}'\mathbf{g}\right],$$

*where $\mathbf{g} \sim \mathcal{N}(\mathbf{0}, \mathbf{I})$ is an $m$-dimensional standard Gaussian random vector.*

# B Omitted Theorems and Proofs

## B.1 Bounding the Estimation Bias

To bound $\|\boldsymbol{\beta}_m^* - \hat{\boldsymbol{\beta}}_m\|_2$, we present a simplified version of Theorem 3.11 in [1] as follows.

**Theorem B.1.** *Under Assumptions A, B, C, D, E, when the sample size $N_m \geq n_m$, with probability at least $\delta_m$,*

$$\|\boldsymbol{\beta}_m^* - \hat{\boldsymbol{\beta}}_m\|_2 \leq \tau_m.$$

The parameters $n_m, \delta_m, \tau_m$ are related to the Gaussian width of the unit ball in $\|\cdot\|_\infty$, the sub-Gaussian norm of $(\mathbf{x}_m, y_m)$, the eigenvalues of the covariance matrix of $(\mathbf{x}_m, y_m)$, as well as the geometric structure of the true regression coefficient $\boldsymbol{\beta}_m^*$. Moreover, $\tau_m$ is decreased as the sample size increases and the uncertainty embedded in $(\mathbf{x}_m, y_m)$ is reduced.

## B.2 Bounding the Distance to the Nearest Neighbors

We will show that the distances between $\mathbf{x}$ and its $K_m$ nearest neighbors could be upper bounded probabilistically. All predictors are assumed to be centered, and independent from each other. In Theorem B.2 we present a lower bound for $\mathbb{P}(\|\mathbf{x} - \mathbf{x}_{m(i)}\|_{\mathbf{W}} \leq \bar{w}_m, \ i = 1, \ldots, K_m)$, for any positive definite diagonal matrix $\mathbf{W}$.

**Theorem B.2.** *Suppose we are given $N_m$ i.i.d. samples $(\mathbf{x}_{mi}, y_{mi})$, $i \in [N_m]$, drawn from some unknown probability distribution with finite fourth moment. Every $\mathbf{x}_{mi}$ has independent, centered coordinates:*

$$\mathbb{E}(\mathbf{x}_{mi}) = \mathbf{0}, \ cov(\mathbf{x}_{mi}) = diag\left(\sigma_{m1}^2, \ldots, \sigma_{mp}^2\right), \forall i \in [N_m].$$

*For a fixed predictor $\mathbf{x}$, and any given positive definite diagonal matrix $\mathbf{W} \in \mathbb{R}^{p \times p}$ with diagonal elements $w_j$, $j \in [p]$, and $|w_j| \leq \bar{B}^2$, suppose:*

$$|(x_{mij} - x_j)^2 - (\sigma_{mj}^2 + x_j^2)| \leq T_m, \ a.s., \ \forall i \in [N_m], \ j \in [p],$$

*where $x_{mij}, x_j$ are the $j$-th components of $\mathbf{x}_{mi}$ and $\mathbf{x}$, respectively. Under the condition that $\bar{w}_m^2 > \bar{B}^2 \sum_{j=1}^p (\sigma_{mj}^2 + x_j^2)$, with probability at least $1 - I_{1-p_{m0}}(N_m - K_m + 1, K_m)$,*

$$\|\mathbf{x} - \mathbf{x}_{m(i)}\|_{\mathbf{W}} \leq \bar{w}_m, \ i \in [K_m],$$

*where*

$$I_{1-p_{m0}}(N_m - K_m + 1, K_m) \triangleq \frac{N_m!}{(K_m - 1)!(N_m - K_m)!} \int_0^{1-p_{m0}} t^{N_m - K_m}(1-t)^{K_m - 1} dt,$$

$$p_{m0} = 1 - \exp\left(-\frac{\sigma_m^2}{T_m^2} g\left(\frac{T_m\left(\bar{w}_m^2/\bar{B}^2 - \sum_j(\sigma_{mj}^2 + x_j^2)\right)}{\sigma_m^2}\right)\right),$$

*and*

$$\sigma_m = \sqrt{\sum_{j=1}^p var\left((x_{mij} - x_j)^2\right)}, \qquad g(u) = (1+u)\log(1+u) - u.$$

*Proof.* To simplify the notation, we will omit the subscript $m$ in all proofs, e.g., using $\mathbf{x}_i$ and $\mathbf{x}_{(i)}$ for $\mathbf{x}_{mi}$ and $\mathbf{x}_{m(i)}$, respectively, and $N$ for $N_m$. Define the event $\mathcal{A}_i := \{\|\mathbf{x}_i - \mathbf{x}\|_{\bar{B}^2 \mathbf{I}} \leq \bar{w}\}$. As long as we can calculate the probability that at least $K$ of $\mathcal{A}_i$, $i \in [N]$, occur, we are able to provide a lower bound on $\mathbb{P}(\|\mathbf{x} - \mathbf{x}_{(i)}\|_{\mathbf{W}} \leq \bar{w}, \ i \in [K])$. Note that given $\mathbf{x}$, $\mathcal{A}_i$, $i \in [N]$, are independent

and equiprobable, since $\mathbf{x}_i$, $i \in [N]$, are i.i.d. Based on Bennett's inequality [5], we have:

$$
\begin{aligned}
\mathbb{P}(\mathcal{A}_i) &= \mathbb{P}(\|\mathbf{x}_i - \mathbf{x}\|_{\bar{B}^2 \mathbf{I}}^2 \le \bar{w}^2) \\
&= \mathbb{P}\Big( \bar{B}^2 (x_{i1} - x_1)^2 + \ldots + \bar{B}^2 (x_{ip} - x_p)^2 \le \bar{w}^2 \Big) \\
&= \mathbb{P}(t_1 + \cdots + t_p \le \bar{w}^2 / \bar{B}^2) \\
&= \mathbb{P}\bigg( \sum_j \Big( t_j - (\sigma_j^2 + x_j^2) \Big) \le \bar{w}^2/\bar{B}^2 - \sum_j (\sigma_j^2 + x_j^2) \bigg) \\
&\ge 1 - \exp\bigg( -\frac{\sigma^2}{T^2} g\Big( \frac{T\big( \bar{w}^2/\bar{B}^2 - \sum_j (\sigma_j^2 + x_j^2) \big)}{\sigma^2} \Big) \bigg) \\
&\triangleq p_0,
\end{aligned}
$$

where $t_j = (x_{ij} - x_j)^2$, $j \in [p]$; $\sigma^2 = \sum_j \mathrm{var}(t_j)$. In the above derivation, we used the fact that $t_j$, $j \in [p]$, are independent, and $|t_j - \mathbb{E}[t_j]| \le T$, a.s., $\forall j$.

Given the lower bound for $\mathbb{P}(\mathcal{A}_i)$, we can derive a lower bound for the probability that exactly $K$ of $\mathcal{A}_i$, $i \in [N]$, occur. For a given $\mathbf{x}$, $\mathcal{A}_i$, $i \in [N]$, are independent, and thus,

$$
\begin{aligned}
\mathbb{P}(\|\mathbf{x} - \mathbf{x}_{(i)}\|_{\mathbf{W}} \le \bar{w}, \ i \in [K]) &\ge \mathbb{P}(\text{at least } K \text{ of } \mathcal{A}_i, i \in [N] \text{ occur}) \\
&= \sum_{k=K}^N \binom{N}{k} \Big( \mathbb{P}(\mathcal{A}_i) \Big)^k \Big( 1 - \mathbb{P}(\mathcal{A}_i) \Big)^{N-k} \\
&\ge \sum_{k=K}^N \binom{N}{k} p_0^k (1 - p_0)^{N-k} \\
&= 1 - I_{1-p_0}(N - K + 1, K),
\end{aligned}
$$

where $I_{1-p_0}(N - K + 1, K)$ is the *regularized incomplete beta function* defined as $I_{1-p_0}(N - K + 1, K) \triangleq (N - K + 1)\binom{N}{K-1} \int_0^{1-p_0} t^{N-K}(1-t)^{K-1} dt$. The bound above used the monotonicity of the binomial tail distribution in the "success" probability. $\square$

## B.3 Proof of Theorem 2.1

*Proof.* We omit the subscript $m$ for simplicity. By Theorems B.1 and B.2, we have

$$
\begin{aligned}
|(\mathbf{x} - \mathbf{x}_{(i)})'(\boldsymbol{\beta}^* - \hat{\boldsymbol{\beta}})| &= |(\mathbf{x} - \mathbf{x}_{(i)})'\hat{\mathbf{W}}^{\frac{1}{2}} \hat{\mathbf{W}}^{-\frac{1}{2}} (\boldsymbol{\beta}^* - \hat{\boldsymbol{\beta}})| \\
&\le \|(\mathbf{x} - \mathbf{x}_{(i)})'\hat{\mathbf{W}}^{\frac{1}{2}}\|_2 \|\hat{\mathbf{W}}^{-\frac{1}{2}} (\boldsymbol{\beta}^* - \hat{\boldsymbol{\beta}})\|_2 \\
&\le \frac{\bar{w}\tau}{b},
\end{aligned}
$$

where the second inequality used the fact that $\|\hat{\mathbf{W}}^{-\frac{1}{2}} (\boldsymbol{\beta}^* - \hat{\boldsymbol{\beta}})\|_2 \le \frac{\tau}{b}$ if $\|\boldsymbol{\beta}^* - \hat{\boldsymbol{\beta}}\|_2 \le \tau$, which can be verified by the Courant-Fischer Theorem, and the fact that $\hat{\mathbf{W}}$ is diagonal with elements $\hat{\beta}_1^2, \ldots, \hat{\beta}_p^2$, and $|\hat{\beta}_j| \ge b$. Based on the inequality $\big( \sum_{i=1}^n a_i \big)^2 \le n \big( \sum_{i=1}^n a_i^2 \big)$, we know:

$$
\begin{aligned}
|(\mathbf{x} - \mathbf{x}_{(i)})'\hat{\boldsymbol{\beta}}| &= \Big| \sum_{j=1}^p \hat{\beta}_j (\mathbf{x} - \mathbf{x}_{(i)})_j \Big| \\
&\le \sqrt{ p \sum_{j=1}^p \Big( \hat{\beta}_j (\mathbf{x} - \mathbf{x}_{(i)})_j \Big)^2 } \\
&= \sqrt{ p(\mathbf{x} - \mathbf{x}_{(i)})' \hat{\mathbf{W}} (\mathbf{x} - \mathbf{x}_{(i)}) } \\
&\le \sqrt{p}\bar{w}.
\end{aligned}
$$

Therefore,

$$
\begin{aligned}
|(\mathbf{x} - \mathbf{x}_{(i)})'\boldsymbol{\beta}^*| &= |(\mathbf{x} - \mathbf{x}_{(i)})'(\boldsymbol{\beta}^* - \hat{\boldsymbol{\beta}}) + (\mathbf{x} - \mathbf{x}_{(i)})'\hat{\boldsymbol{\beta}}| \\
&\leq |(\mathbf{x} - \mathbf{x}_{(i)})'(\boldsymbol{\beta}^* - \hat{\boldsymbol{\beta}})| + |(\mathbf{x} - \mathbf{x}_{(i)})'\hat{\boldsymbol{\beta}}| \\
&\leq \frac{\bar{w}\tau}{b} + \sqrt{p}\bar{w}.
\end{aligned}
$$

Thus, for a given $\mathbf{x}$,

$$
\begin{aligned}
&\mathbb{E}\Big[(\hat{y}(\mathbf{x}) - y(\mathbf{x}))^2 \Big| \mathbf{x}, \mathbf{x}_i\Big] \\
&= \left(\frac{1}{K}\sum_{i=1}^{K}\big((\mathbf{x} - \mathbf{x}_{(i)})'\boldsymbol{\beta}^* + h(\mathbf{x}) - h(\mathbf{x}_{(i)})\big)\right)^2 + \frac{\eta^2}{K} + \eta^2 \\
&\leq \left(\frac{1}{K}\sum_{i=1}^{K}\big(|(\mathbf{x} - \mathbf{x}_{(i)})'\boldsymbol{\beta}^*| + |h(\mathbf{x}) - h(\mathbf{x}_{(i)})|\big)\right)^2 + \frac{\eta^2}{K} + \eta^2 \\
&\leq \left(\frac{\bar{w}\tau}{b} + \sqrt{p}\bar{w} + \frac{L\bar{w}}{\bar{B}}\right)^2 + \frac{\eta^2}{K} + \eta^2
\end{aligned}
\tag{2}
$$

The above bound used both Thms. B.1 and B.2, whose statements hold with probabilities no less than $\delta$ and $1 - I_{1-p_0}(N - K + 1, K)$ w.r.t. sampling, respectively. Let $\mathcal{A}$ and $\mathcal{B}$ the events corresponding to the statements of Thms. B.1 and B.2 being satisfied. Using bar to denote complement, and the union bound, it follows that (2) holds with probability

$$
\mathbb{P}(\mathcal{A} \cap \mathcal{B}) = 1 - \mathbb{P}(\overline{\mathcal{A} \cap \mathcal{B}}) = 1 - \mathbb{P}(\bar{\mathcal{A}} \cup \bar{\mathcal{B}}) \geq \delta - I_{1-p_0}(N - K + 1, K).
$$

The probability bound can be easily derived using Markov's inequality. $\qquad\square$

## B.4 Proof of Theorem 3.1

*Proof.* The proof borrows ideas from Theorem 1.5 in [2]. Define $W_m \triangleq e^{-\xi\hat{y}_m(\mathbf{x})} / \sum_{j=1}^{M} e^{-\xi\hat{y}_j(\mathbf{x})}$, and $\phi \triangleq \sum_{m=1}^{M} e^{-\xi\hat{y}_m(\mathbf{x})} e^{-\xi y_m(\mathbf{x})}$. Then,

$$
\begin{aligned}
\phi &= \left(\sum_{j=1}^{M} e^{-\xi\hat{y}_j(\mathbf{x})}\right) \sum_{m=1}^{M} W_m e^{-\xi y_m(\mathbf{x})} \\
&\leq \left(\sum_{j=1}^{M} e^{-\xi\hat{y}_j(\mathbf{x})}\right) \sum_{m=1}^{M} W_m\big(1 - \xi y_m(\mathbf{x}) + \xi^2 y_m^2(\mathbf{x})\big) \\
&= \left(\sum_{j=1}^{M} e^{-\xi\hat{y}_j(\mathbf{x})}\right)\left(1 - \xi \sum_{m=1}^{M} W_m y_m(\mathbf{x}) + \xi^2 \sum_{m=1}^{M} W_m y_m^2(\mathbf{x})\right) \\
&\leq \left(\sum_{j=1}^{M} e^{-\xi\hat{y}_j(\mathbf{x})}\right) e^{-\xi \sum_{m=1}^{M} W_m y_m(\mathbf{x}) + \xi^2 \sum_{m=1}^{M} W_m y_m^2(\mathbf{x})},
\end{aligned}
$$

where the first inequality uses the fact that for $x \geq 0$, $e^{-x} \leq 1 - x + x^2$, and the last inequality is due to the fact that $1 + x \leq e^x$. Next let us examine the sum of exponentials:

$$
\begin{aligned}
\sum_{j=1}^{M} e^{-\xi\hat{y}_j(\mathbf{x})} &\leq \sum_{j=1}^{M}\left(1 - \xi\hat{y}_j(\mathbf{x}) + \xi^2\hat{y}_j^2(\mathbf{x})\right) \\
&= M\left(1 - \xi\frac{1}{M}\sum_{j=1}^{M}\hat{y}_j(\mathbf{x}) + \xi^2\frac{1}{M}\sum_{j=1}^{M}\hat{y}_j^2(\mathbf{x})\right) \\
&\leq M e^{-\xi\frac{1}{M}\sum_{j=1}^{M}\hat{y}_j(\mathbf{x}) + \xi^2\frac{1}{M}\sum_{j=1}^{M}\hat{y}_j^2(\mathbf{x})}.
\end{aligned}
$$

Using the two bounds above, for any $k \in [M]$, we have

$$
\begin{aligned}
e^{-\xi\hat{y}_k(\mathbf{x}) - \xi y_k(\mathbf{x})} &\leq \phi \\
&\leq M e^{-\frac{\xi\sum_{j=1}^{M}\hat{y}_j(\mathbf{x})}{M} + \frac{\xi^2\sum_{j=1}^{M}\hat{y}_j^2(\mathbf{x})}{M} - \xi\sum_{m=1}^{M} W_m y_m(\mathbf{x}) + \xi^2\sum_{m=1}^{M} W_m y_m^2(\mathbf{x})}.
\end{aligned}
\tag{3}
$$

Taking the logarithm on both sides of (3) and dividing by $\xi$, we obtain

$$\frac{1}{M}\sum_{m=1}^{M}\hat{y}_m(\mathbf{x}) + \sum_{m=1}^{M}\frac{e^{-\xi\hat{y}_m(\mathbf{x})}}{\sum_j e^{-\xi\hat{y}_j(\mathbf{x})}}y_m(\mathbf{x}) \le \hat{y}_k(\mathbf{x}) + y_k(\mathbf{x})$$

$$+ \xi\left(\frac{1}{M}\sum_{m=1}^{M}\hat{y}_m^2(\mathbf{x}) + \sum_{m=1}^{M}\frac{e^{-\xi\hat{y}_m(\mathbf{x})}}{\sum_j e^{-\xi\hat{y}_j(\mathbf{x})}}y_m^2(\mathbf{x})\right) + \frac{\log M}{\xi}.$$

$\square$

## B.5 Proof of Theorem 3.2

*Proof.* By the sub-Gaussian assumption we have:

$$\mathbb{P}\left(\sum_k \frac{e^{-\xi\hat{y}_k(\mathbf{x})}}{\sum_j e^{-\xi\hat{y}_j(\mathbf{x})}}\hat{y}_k(\mathbf{x}) > x_{\text{co}} - T(\mathbf{x})\right) \le \mathbb{P}\left(\max_k \hat{y}_k(\mathbf{x}) > x_{\text{co}} - T(\mathbf{x})\right)$$

$$= \mathbb{P}\left(\bigcup_k \{\hat{y}_k(\mathbf{x}) > x_{\text{co}} - T(\mathbf{x})\}\right)$$

$$\le \sum_k \mathbb{P}\left(\hat{y}_k(\mathbf{x}) > x_{\text{co}} - T(\mathbf{x})\right) \quad (4)$$

$$\le \sum_k \exp\left(-\frac{\left(x_{\text{co}} - T(\mathbf{x}) - \mu_{\hat{y}_k}(\mathbf{x})\right)^2}{2C_k^2(\mathbf{x})}\right).$$

Note that the probability in (4) is taken with respect to the measure of the training samples. We essentially want to find the largest threshold $T(\mathbf{x})$ such that the probability of the expected improvement being less than $T(\mathbf{x})$ is small. Given a small $0 < \bar{\epsilon} < 1$ and due to (4), to satisfy

$$\mathbb{P}\left(\sum_k \frac{e^{-\xi\hat{y}_k(\mathbf{x})}}{\sum_j e^{-\xi\hat{y}_j(\mathbf{x})}}\hat{y}_k(\mathbf{x}) > x_{\text{co}} - T(\mathbf{x})\right) \le \bar{\epsilon},$$

it suffices to set:

$$\sum_k \exp\left(-\frac{\left(x_{\text{co}} - T(\mathbf{x}) - \mu_{\hat{y}_k}(\mathbf{x})\right)^2}{2C_k^2(\mathbf{x})}\right) \le \bar{\epsilon}. \quad (5)$$

A sufficient condition for (5) is:

$$\exp\left(-\frac{\left(x_{\text{co}} - T(\mathbf{x}) - \mu_{\hat{y}_m}(\mathbf{x})\right)^2}{2C_m^2(\mathbf{x})}\right) \le \frac{\bar{\epsilon}}{M}, \quad \forall m \in [M],$$

which yields that,

$$T(\mathbf{x}) \le x_{\text{co}} - \mu_{\hat{y}_m}(\mathbf{x}) - \sqrt{-2C_m^2(\mathbf{x})\log(\bar{\epsilon}/M)}, \ \forall m \in [M]. \quad (6)$$

Given that $T(\mathbf{x})$ is non-negative, we set the largest possible threshold satisfying (6) to:

$$T(\mathbf{x}) = \max\left(0, \ \min_m\left(x_{\text{co}} - \mu_{\hat{y}_m}(\mathbf{x}) - \sqrt{-2C_m^2(\mathbf{x})\log(\bar{\epsilon}/M)}\right)\right).$$

When using a deterministic policy ($\xi \to \infty$), for any $m \in [M]$, we have

$$\mathbb{P}(\min_m \hat{y}_m(\mathbf{x}) > x_{\text{co}} - T(\mathbf{x})) = \mathbb{P}\left(\bigcap_m \{\hat{y}_m(\mathbf{x}) > x_{\text{co}} - T(\mathbf{x})\}\right)$$

$$\le \mathbb{P}(\hat{y}_m(\mathbf{x}) > x_{\text{co}} - T(\mathbf{x}))$$

$$\le \exp\left(-\frac{\left(x_{\text{co}} - T(\mathbf{x}) - \mu_{\hat{y}_m}(\mathbf{x})\right)^2}{2C_m^2(\mathbf{x})}\right).$$

Similarly, to make

$$\mathbb{P}\left(\min_m \hat{y}_m(\mathbf{x}) > x_{\text{co}} - T(\mathbf{x})\right) \le \bar{\epsilon},$$

we set:

$$T(\mathbf{x}) = \max\left(0, \ \min_m\left(x_{\text{co}} - \mu_{\hat{y}_m}(\mathbf{x}) - \sqrt{-2C_m^2(\mathbf{x})\log\bar{\epsilon}}\right)\right),$$

which establishes the desired result. $\square$

## C Numerical Experiments Details

### C.1 Cohort Selection

The patients that meet the following criteria are included in the hypertension dataset:

- Patients present in the system for at least 1 year;
- Received at least one type of cardiovascular medications, including ACE inhibitors, Angiotensin Receptor Blockers (ARB), calcium channel blockers, diuretics, $\alpha$-blockers and $\beta$-blockers, and had at least one medical record 10 days before this prescription.
- Had at least one recorded diagnosis of hypertension (corresponding to the ICD-9 diagnosis codes 401-405);
- Had at least three measurements of the systolic blood pressure.

### C.2 Predictive Performance of Various Models

We use four metrics to evaluate the predictive power of various models on the test set:

- The R-square:

$$\mathrm{R}^2(\mathbf{y}, \hat{\mathbf{y}}) = 1 - \frac{\sum_{i=1}^{N_t}(y_i - \hat{y}_i)^2}{\sum_{i=1}^{N_t}(y_i - \bar{y})^2},$$

    where $\mathbf{y} = (y_1, \ldots, y_{N_t})$ and $\hat{\mathbf{y}} = (\hat{y}_1, \ldots, \hat{y}_{N_t})$ are the vectors of the true (observed) and predicted outcomes, respectively, with $N_t$ the size of the test set, and $\bar{y} = (1/N_t)\sum_{i=1}^{N_t} y_i$.

- The *Mean Squared Error (MSE)*:

$$\mathrm{MSE}(\mathbf{y}, \hat{\mathbf{y}}) = \frac{1}{N_t}\sum_{i=1}^{N_t}(y_i - \hat{y}_i)^2.$$

- The *Mean Absolute Error (MeanAE)* that is more robust to large deviations than the MSE since the absolute value function increases more slowly than the square function over large (absolute) values of the argument.

$$\mathrm{MeanAE}(\mathbf{y}, \hat{\mathbf{y}}) = \frac{1}{N_t}\sum_{i=1}^{N_t}|y_i - \hat{y}_i|.$$

- The MedianAE which can be viewed as a robust measure of the MeanAE, computing the median of the absolute deviations:

$$\mathrm{MedianAE}(\mathbf{y}, \hat{\mathbf{y}}) = \mathrm{Median}\left(|y_i - \hat{y}_i|, i = 1, \ldots, N_t\right).$$

The out-of-sample performance metrics of the various models on the hypertension dataset are shown in Table 1, where the numbers in the parentheses show the improvement of DRLR informed K-NN compared against other methods. Huber refers to the robust regression method proposed in [3, 4], and CART refers to the *Classification And Regression Trees*. Huber/OLS/LASSO + K-NN means fitting a K-NN regression model with a Huber/OLS/LASSO-weighted distance metric. We note that in order to produce well-defined and meaningful predictive performance metrics, the dataset used to generate Table 1 did not group the patients by their prescriptions. A universal model was fit to all patients using the prescription as one of the predictors. Nevertheless, it would still be considered as a fair comparison as all models were evaluated on the same dataset. The results provide supporting evidence for the validity of our DRLR+K-NN model that outperforms all others in all metrics, and is thus used for predicting the outcomes of counterfactual treatments.

## Footnotes

*http://sites.bu.edu/paschalidis

## References

[1] Ruidi Chen and Ioannis Ch Paschalidis. A robust learning approach for regression models based on distributionally robust optimization. *Journal of Machine Learning Research*, 19(13), 2018.

Table 1: Performance of different models for predicting future systolic blood pressure for hypertension patients.

| Methods | $R^2$ | MSE | MeanAE | MedianAE |
|---|---|---|---|---|
| OLS | 0.31 (14%) | 170.80 (6%) | 10.09 (7%) | 8.15 (9%) |
| LASSO | 0.31 (14%) | 170.83 (6%) | 10.08 (7%) | 8.22 (10%) |
| Huber | 0.22 (62%) | 193.54 (17%) | 10.70 (12%) | 8.61 (14%) |
| RLAD | 0.30 (18%) | 173.32 (8%) | 10.11 (7%) | 8.28 (11%) |
| K-NN | 0.33 (10%) | 167.41 (5%) | 9.62 (2%) | 7.50 (2%) |
| OLS+K-NN | 0.35 (1%) | 160.22 (0%) | 9.42 (0%) | 7.49 (1%) |
| LASSO+K-NN | 0.32 (12%) | 169.50 (6%) | 9.74 (3%) | 7.73 (5%) |
| Huber+K-NN | 0.32 (10%) | 167.92 (5%) | 9.71 (3%) | 7.84 (6%) |
| DRLR+K-NN | 0.36 (N/A) | 159.74 (N/A) | 9.42 (N/A) | 7.38 (N/A) |
| CART | 0.25 (43%) | 186.23 (14%) | 10.34 (9%) | 8.22 (10%) |

Table 2: Feature importance from the DRLR model for the hypertension dataset.

| Features | Regression coefficients |
|---|---|
| measurement: systolic blood pressure | 7.62 |
| age | 1.87 |
| lab test: sodium | 1.29 |
| lab test: hemoglobin | 1.26 |
| prescription: calcium channel blockers | 0.98 |
| lab test: blood glucose | 0.93 |
| lab test: hematocrit | -0.82 |
| sex: female | 0.76 |
| lab:mean corpuscular volume | -0.61 |
| diagnosis: asthma | -0.61 |
| prescription: ARB | 0.57 |
| diagnosis: cataract | 0.57 |
| diagnosis: chronic ischemic heart disease | -0.56 |
| lab test: potassium | 0.55 |
| diagnosis: heart failure | -0.53 |
| prescription: diuretics | 0.53 |
| diagnosis: cardiac dysrhythmias | -0.51 |
| diagnosis obesity | 0.46 |
| race: Caucasian | -0.46 |
| diagnosis: disorders of fluid electrolyte and acid-base balance | 0.45 |

[2] Elad Hazan. Introduction to online convex optimization. *Foundations and Trends® in Optimization*, 2(3-4):157–325, 2016.

[3] Peter J Huber. Robust estimation of a location parameter. *The Annals of Mathematical Statistics*, pages 73–101, 1964.

[4] Peter J Huber. Robust regression: asymptotics, conjectures and monte carlo. *The Annals of Statistics*, pages 799–821, 1973.

[5] Roman Vershynin. *High-dimensional probability: An introduction with applications in data science*. Cambridge University Press (to appear), 2017.