[Reviews · NeurIPS 2019]

Reviewer 1



prediction-based prescriptive method is developed in the paper

Reviewer 2



Major concerns: The data described from the paper are with labeling (response y_m) for each action m \in M. Therefore it seems to me the counterfactual effects can also be predicted by just using the regression based method (either parametric or nonparametric). The KNN method is more of an unsupervised method to do a re-weighting, than a nonparametric component in the model. I am not fully convinced of the benefits of using KNN. On the other hand, I see potential problems: for example, this method could be sensitive to the choice of number of neighbors K; it treats all the neighbors with the same weight in the final response prediction.

Reviewer 3



This paper has potential, but a significant rewrite is needed to discuss the related literature and clarify the results in context.

Reviewer 4



The paper tackles the problem of predicting the outcome of an action chosen from a set of possible actions, The outcome is a function of the action, having a linear component, non-linear component and some additive noise. The idea is first finding a linear function minimizing the deviation from the outcomes, for every distribution which is "close" to the empirical distribution (by the Wasserstein distance). Idea which was analyzed before. The idea added in the paper is using the resulting linear-regression coefficient to build a metric upon samples from the same group and then produce prediction which is the average of the outcomes for the K-nearest neighbors. This way the prediction can leverage not only the private history of the specific instance but also the outcomes of "close" instances. The paper provide bound on the resulting error under several assumptions. The idea is interesting and seems novel. The analysis isn't trivial. The paper is written in a relative clear manner.

Reviewer 5



Originality - The work nicely combines two methods to calculate an estimator with good prediction. However, it is not clear from the paper why this combination of methods provides the optimal performance. Quality - The claimes are well suported by theoritical analysis and experimntal results. Clarity - The submission is clearly written. Significance - This paper offers an interesting approach for feature selecting for knn regression, based on weighted metric. This is an interesting research direction, on the theoretical and applicative areas.

[Author Response · NeurIPS 2019]

1. **Our contribution.** The DRLR informed KNN builds on a previously proposed method using OLS to inform KNN. Yet, our key, differentiating contributions lie in that: ($i$) the use of DRLR induces a more robust identification of the important factors that affect the future response and, thus, the neighbors being selected provide a better representation of the individual in query in the presence of outliers; ($ii$) we propose a randomized prescriptive policy that can potentially correct for the prediction bias and thus adds further robustness; ($iii$) in contrast to the earlier work, we establish new, rigorous theoretical results on the predictive and prescriptive performance of the proposed method; and ($iv$) we derive a closed-form expression for the threshold level that is used to activate the prescriptive rule.

2. **The benefit of KNN.** If we just use DRLR (which is a linear regression model) to predict the counterfactuals, the prediction will be far from accurate since the non-linearity in the data is not well explained. The KNN model builds locally linear (and globally nonlinear) predictions using information from neighbors, accounting for the non-linearity that is not captured by DRLR. Furthermore, it is nonparametric, easy to estimate and is efficient to solve. In consideration of its sensitivity to the choice of the number of neighbors, $K$, we proposed a refinement of the policy described in the end of the experimental section, where a patient-specific $K$ was used, discarding the neighbors that are relatively far away from the patient in query. In the final response prediction, we treat all the neighbors equally, using the same weight. An alternative is to take a distance-weighted average of the responses of neighbors; we tried this on our medical datasets, but we found that its effect is not significantly different from the strategy where a uniform average of the responses is taken. Notice that we discarded the neighbors that are relatively far away from the individual in the refined policy, which can be considered as a weighted average of the responses. We also want to point out that the theoretical analysis can be easily adapted to the weighted average response prediction.

3. **Using DRLR based weights to inform KNN.** The intuition for using a weighted distance metric in KNN is to amplify the weight of features that are most predictive of the future response and downweight the unimportant ones. We use DRLR to inform KNN because it produces a robust identification of the important features. As a result, the selected samples are close to the individual in the most relevant features, and their corresponding response values should serve as a good approximation. Combining DRLR with KNN produces predictions that account for the non-linearity in the data and are robust to outliers. It is hard to define "optimal performance" for our proposed method. What we can show is that its prediction bias depends on the accuracy of the linear coefficient estimate, the similarity between the individual in query and its K-nearest neighbors, the dimensionality of data, and the smoothness of the regression hypothesis. Robustness and nonlinearity are the focus of this work, which are respectively taken care of by the DRLR and KNN models.

4. **Metric regression.** Our method constructs a locally linear estimator of the future outcome through learning a robust metric in the feature space. Different from the classical metric learning works (e.g., Gottlieb et al. [2017]), we solve a downstream decision making problem by utilizing the information filtered by the learned metric. Gottlieb et al. [2017] focuses on the computational aspect of solving the metric regression problem. They significantly improve the computational efficiency compared to solving a convex program. By contrast, we focus on developing a novel method for the optimal decision making problem rather than improving the algorithmic efficiency. A direct comparison of the two papers easily shows how different they are. Moreover, Gottlieb et al. [2017] studies only the regression problem, whereas we considered a richer framework of combining regression with a randomized prescriptive policy. For the generalization bounds, we offered similar insights to their results. Theorem 5.1 in their paper provided a risk bound that depends on the empirical risk (reflected in $\tau_m$ and $\bar{w}_m$ of our bound), the dimensionality of data (p), and the smoothness of the regression hypothesis ($L_m$). Their result also considered the runtime-precision tradeoff which we did not take into account. We would be happy to update our literature review to make these connections.

5. **The notation.** We apologize for the confusion in the notations. Due to the page limit, we cannot present the intermediate results that lead to the main theorem, and many notations coming from the intermediate theorems were not clearly stated (these are included in the supplementary file). We will fix this in the revision, and be more clear about the notation and assumptions. The rate with respect to the sample size $N$ is $1/N^2$, since $\tau_m$ is proportional to $1/N$.

# References

Lee-Ad Gottlieb, Aryeh Kontorovich, and Robert Krauthgamer. Efficient regression in metric spaces via approximate lipschitz extension. *IEEE Transactions on Information Theory*, 63(8):4838–4849, 2017.


[Meta-Review · NeurIPS 2019]

The paper addresses the problem of predicting the outcome of an action chosen from a set of possible actions with Distributionally Robust Nearest-Neighbor Regression. Additionally to the description of the method and its theoretical analysis, an application to finding optimal prescriptions for patients with hypertension is studied. The reviewers found that the paper was written in a clear manner. The ideas of the paper were found interesting and novel. The work brings a non trivial theoretical analysis. Some concerns were raised about 1), the benefit of using k-NN instead of sole Distributionally Robust Linear Regression 2), assumptions in the theoretical analysis and 3), the lack of discussion with a seemingly related literature in metric regression. The rebuttal satisfied most of the objections -- the authors are strongly encouraged to follow through, including the additional literature review they agreed to include. Assuming these are implemented as stated, this paper will be suitable for publication. This meta-review was reviewed and revised by the Program Chairs, based on discussions with the Senior Area Chair.